# Umbilical Endometriosis: A Systematic Literature Review and Pathogenic Theory Proposal

**DOI:** 10.3390/jcm11040995

**Published:** 2022-02-14

**Authors:** Dhouha Dridi, Francesca Chiaffarino, Fabio Parazzini, Agnese Donati, Laura Buggio, Massimiliano Brambilla, Giorgio Alberto Croci, Paolo Vercellini

**Affiliations:** 1Gynecology Unit, Fondazione IRCCS Ca’ Granda Ospedale Maggiore Policlinico, 20122 Milan, Italy; francesca.chiaffarino@gmail.com (F.C.); agnese.do@tiscali.it (A.D.); buggiolaura@gmail.com (L.B.); paolo.vercellini@unimi.it (P.V.); 2Department of Clinical Sciences and Community Health, University of Milan, 20122 Milan, Italy; fabio.parazzini@unimi.it; 3Plastic Surgery Service, Gynecology Unit, Fondazione IRCCS Ca’ Granda Ospedale Maggiore Policlinico, 20122 Milan, Italy; massimiliano.brambilla@policlinico.mi.it; 4Division of Pathology, Fondazione IRCCS Ca’ Granda Ospedale Maggiore Policlinico, 20122 Milan, Italy; giorgio.croci@unimi.it; 5Department of Pathophysiology and Transplantation, University of Milan, 20122 Milan, Italy

**Keywords:** endometriosis, umbilical endometriosis, Villar’s nodule, symptoms, pain, frequency

## Abstract

Umbilical endometriosis represents 30–40% of abdominal wall endometriosis and around 0.5–1.0% of all cases of endometriosis. The aim of this systematic review is to revisit the epidemiology, signs, and symptoms and to formulate a pathogenic theory based on literature data. We performed a systematic literature review using the PubMed and Embase databases from 1 January 1950 to 7 February 2021, according to the PRISMA guidelines. The review was registered at PROSPERO (CRD42021239670). Studies were selected if they reported original data on umbilical endometriosis nodule defined at histopathological examination and described as the presence of endometrial glands and/or stromal cells in the connective tissue. A total of 11 studies (10 retrospective and one prospective), and 14 case series were included in the present review. Overall, 232 umbilical endometriosis cases were reported, with the number per study ranging from 1 to 96. Umbilical endometriosis was observed in 76 (20.9%; 95% CI 17.1–25.4) of the women included in studies reporting information on the total number of cases of abdominal wall endometriosis. Umbilical endometriosis was considered a primary form in 68.4% (158/231, 95% CI 62.1–74.1) of cases. A history of endometriosis and previous abdominal surgery were reported in 37.9% (25/66, 95% CI 27.2–49.9) and 31.0% (72/232, 95% CI 25.4–37.3) of cases, respectively. Pain was described in 83% of the women (137/165, 95% CI 76.6–88.0), followed by catamenial symptoms in 83.5% (142/170, 95% CI, 77.2–88.4) and bleeding in 50.9% (89/175, 95% CI 43.5–58.2). In the 148 women followed for a period ranging from three to 92.5 months, seven (4.7%, 95% CI 2.3–9.4) recurrences were observed. The results of this analysis show that umbilical endometriosis represents about 20% of all the abdominal wall endometriotic lesions and that over two thirds of cases are primary umbilical endometriosis forms. Pain and catamenial symptoms are the most common complaints that suggest the diagnosis. Primary umbilical endometriosis may originate from implantation of regurgitated endometrial cells conveyed by the clockwise peritoneal circulation up to the right hemidiaphragm and funneled toward the umbilicus by the falciform and round liver ligaments.

## 1. Introduction

Endometriosis is a benign gynecological disorder that affects about 5% of reproductive-aged women [1,2,3,4]. The pelvic cavity is the most common location of endometriotic implants, but about 12% of lesions are extragenital [5,6] and, among the extra-pelvic sites, endometriosis of the abdominal wall (AWE) is the most common [7]. Umbilical endometriosis (UE), or Villar’s nodule, as first described by Villar in 1886, is defined as the presence of endometrial glands and/or stroma within the umbilicus. It is a rare form of endometriosis with a frequency of around 0.4–4% of extragenital lesions [8,9] and around 0.5–1% of all cases of endometriosis [10,11,12], and it has been reported that it represents 30–40% of cases of abdominal wall endometriosis (AWE) [13]. Generally, UE presents with a red, purple, or black umbilical nodule with a diameter ranging from 0.5 to 3 cm [14]. According to Hirata et al. (2020), the risk of malignant transformation of UE is about 3% [9].

Two types of UE have been described. Primary UE occurs in the absence of a surgical history. Severe theories have been proposed, such as the migration of endometrial cells through the abdominal cavity, the lymphatic system, or embryonic remnants in the umbilical fold (e.g., the urachus and umbilical vessels); genetic predisposition; and immunologic defects [14]. Romera-Barba et al., demonstrated that the disease occurs after prolonged exposure to the metaplastic and environmental factors [15], whereas secondary UE arises on scar tissue following abdominal procedures such as laparoscopy [9,14]. The distinction between primary and secondary development of UE appears to be important for the understanding of the pathogenic mechanism of this disease form [16].

Hirata et al. [17] suggested guidelines for management of extragenital endometriosis and claims for umbilical endometriosis that radical surgery with wide local excision represents the primary treatment. Surgical excision is strongly recommended but with weak supporting evidence because of unknown long-term efficacy and complications. In contrast, medical treatment is weakly recommended due to limited supporting data and lack of studies comparing medical and surgical treatment for umbilical endometriosis.

Due to the low frequency of the condition, limited data are available on the prevalence of primary and secondary UE and on the associated symptoms. We conducted this systematic review of published studies and case series to define the epidemiological aspects, revisit the signs and symptoms, and suggest a potential pathogenic theory of UE based on anatomic-physiological considerations and the pattern of distribution of associated endometriotic lesions.

## 2. Materials and Methods

The present review was conducted according to the PRISMA guidelines (Appendix A) [18] and was registered at PROSPERO (CRD42021239670).

We performed a systematic literature search using the electronic databases PubMed and Embase on 07 February 2021. The search term “endometriosis” was used as the medical subject heading (MeSH PUBMED) or Embase subject headings (EMTREE EMBASE) term or as free text in combination with the term “umbilical”. The search was limited to full-length articles published in English, French, and Italian since 1 January 1950.

A PICOS (Patient, Intervention, Comparator, Outcome, Study) design structure was used to develop the study questions and the inclusion/exclusion criteria (Appendix A).

Studies were selected if they reported original data on UE nodule defined through histopathological examination and described as the presence of endometrial glands and/or stromal cells in the connective tissue. Studies published from 1950 were identified, with no restriction regarding the age of women. Data presented exclusively as case reports, abstracts in national and international meetings, or review articles that did not include original findings were excluded.

Two authors (D.D. and A.D.) reviewed the papers and independently selected the articles eligible for the systematic review. In the case of more than one publication with the same study population and data, we considered only the more recent article or the one that included the most details. Moreover, bibliographies of the retrieved papers were reviewed to identify other potentially relevant studies. Disagreements were resolved by discussion.

From each publication we extracted the following information: author, year of publication, country, study design, number of patients with UE, number of women with AWE, age of participants, UE classification (primary or secondary), clinical features, previous pregnancy, mode of delivery, history of endometriosis, previous surgery, concomitant endometriosis if pelvic surgery was conducted concomitantly with umbilical nodule’s excision, and rate of postoperative UE recurrences.

Information on the methodological quality of included studies was assessed based on the Methodological Index for Non-Randomised Studies (MINORS), a validated instrument which is designed for assessment of methodological quality of non-randomized studies in surgery [19]. Briefly, the studies were evaluated on eight pre-defined items with a maximum score of 16. The studies with MINORS scores under 6 were excluded.

For the purpose of this analysis, in the tables we have defined as “clinical series” studies that have reported data on the total of AWE, and as “case series” studies that have reported data on UE cases only. Moreover, we have defined as catamenial symptoms the following complaints: swelling, color changes, consistency change, and tenesmus associated with or worsening during menstruation.

For each study with binary outcomes, we calculated the 95% confidence intervals (CI) of the estimated proportion.

## 3. Results

Database searching identified 1096 articles. After removing duplicates, 753 records were screened and 43 were then considered for eligibility. Nine reports were excluded for lack of features; six were excluded because, although not explicitly stated in the title and abstracts, they were case reports. One clinical series [20] and two case series [12,21], although eligible, were excluded for low MINORS scores. Eventually, we selected 10 retrospective studies [22,23,24,25,26,27,28,29,30,31] on recorded cases in a defined period, one prospective study [32], and 14 cases series [9,33,34,35,36,37,38,39,40,41,42,43,44,45]. The flow diagram of the literature search results is shown in Figure 1.

Table 1 shows the study design, number of women with AWE (data available only for clinical series), number of women with UE, mean age of women, primary or secondary origin of reported cases of UE, and MINORS score for each of the 25 papers considered in this review.

The quality of the selected studies was sufficiently good according to the MINORS criteria, with the score being nine or more in 15 out of 25 papers (Appendix A). The number of UE cases reported in the considered papers ranged from 2 to 96, being less than 10 in 21 studies.

Overall, 232 UE cases were described. Considering only the 11 studies reporting information on the number of total cases of AWE (i.e., the “clinical series”), UE was reported in 76 women, corresponding to 20.9% of women with AWE (95% CI 17.1–25.4) (Table 1).

The mean age of women with UE was 37.9 years (weighted mean for the number of cases of the studies) and ranged from 28.5 to 47.5 among the studies. Out of the 231 cases for which the information was available, 158 had primary UE (68.4%, 95% CI 62.1–74.1).

Parity, history of cesarean section, abdominal surgery, history of endometriosis, and concomitant pelvic endometriosis are considered in Table 2.

The proportion of parous women ranged from 20% to 100% among studies. Ninety-nine of the 170 women for which the obstetrical history was available were parae (58.2%, 95% CI 50.7–65.4). The mode of delivery was reported in 94 women and 23 of them (24.5%, 95% CI 16.9–34.1) underwent a cesarean section. A history of abdominal surgery was reported in 31.0% of UE patients (72/232, 95% CI 25.4–37.3). Finally, considering only women for which the information was reported, around one third of women with UE received a previous diagnosis of endometriosis (25/66, 37.9%; 95% CI 27.2–49.9) and, when pelvic surgery was conducted at the same time as umbilical nodule excision, concomitant pelvic endometriosis was observed in 34.8% of the UE cases (24/69, 95%CI 24.6–46.6). Despite the variability due to random fluctuation, these findings were largely consistent in the considered studies.

The clinical features are considered in Table 3. Pain was the most common symptom, being reported in 83% of women (137/165, 95% CI 76.6–88.0), followed by previously defined catamenial symptoms in 83.5% of cases (142/170, 95%CI 77.2–88.4). Bleeding was recorded in 89 of 175 patients (50.9%, 95%CI 43.5–58.2).

Information on the frequency or UE recurrence was detailed in 14 studies (Table 4).

The size of the umbilical nodule was reported only in two clinical series [24,28], while it was cited in all case series except two with a mean of 21.2 ± 7.9 mm (data not reported in Table 4) [9,34].

Overall, these studies included 148 women (range 2–87). The follow-up period ranged from three to 92.5 months. A total of seven (4.7%, 95%CI 2.3–9.4) recurrences after the index surgery were reported. Hirata et al., therefore, concluded that there was no statistically significant difference between the risk of recurrence after excision with or without peritoneum or with medical treatment after surgery [9].

## 4. Discussion

The results of this analysis show that UE represents about 21% of all the AWE cases: Most lesions were primary UE (about 70%); a history of endometriosis was reported in almost one third of women of UE; pelvic lesions co-existed in about 35% of the patients; and pain and catamenial symptoms were the most commonly reported complaints.

Before discussing the results of this analysis, potential limitations should be considered. The selected studies reported large differences in the number of UE cases, with less than 10 cases included in most of the studies. However, effect of random fluctuation aside, the results of this analysis are largely consistent among different studies, particularly among the largest ones.

Although, when planning the systematic literature search, articles published in three languages were potentially considered, eventually the selected papers were almost all published in English, and only two in French [38,39] and one in Italian [34]. Authors may be more prone to publish in an international, English language journal if results are “newer”. Further, studies published before the year 1950 were excluded, as papers were sparse, frequently lacking histological diagnosis, and differed in diagnostic criteria.

The results of this systematic review are consistent with the findings of previous literature reviews. For example, in a review published in 2007 and including 122 patients with documented UE from 1966 to 2007 and 109 cases reported before 1953, the mean age of the study population was 37.7 years, and 27% of cases had a history of endometriosis [8].

Concerning symptoms, Calagna et al. [5] showed that intermittent pain in the umbilical area was the most common complaint reported by 66.0% of the study women, whereas cyclic bleeding from the umbilical site was described by 43.3% of them. Our findings are consistent but slightly higher than their results.

The data emerging from our review show that the post-operative recurrence rate of the UE is very low. In fact, a total of 7/148 (4.7%, 95%CI 2.3–9.4) recurrences were observed during a follow-up period ranging from three to 92.5 months. This suggests that surgery is an effective treatment for this endometriotic lesion type.

An interesting finding of the present review is the fact that primary UE was the most common form; secondary was the iatrogenic form, representing 31.6% of cases. A history of endometriosis was reported by 37.9% of women. In a literature review published in 2015 (Calagna et al.) [5], the authors estimated that about 20% of cases of primary UE had a diagnosis of pelvic endometriosis. In the present review, this figure is substantially higher, as concomitant endometriotic lesions were observed in over one third of women who underwent pelvic visualization. This observation offers some insights in the pathogenesis of UE. Primary endometriosis is obviously the disease form of interest here, as the origin of secondary endometriosis is mainly iatrogenic.

In cases of isolated umbilical endometriosis, the disease might arise from metaplastic changes of urachal remnants [45]. Otherwise, in consideration of the frequency of UE among cases of AWE (21% in our review), it can be hypothesized that endometrial cells may stem from the umbilical cord at birth. According to Calagna et al. (2015) [5], inflammation of the tissues around endometriotic pelvic implant may favor the shedding of endometriotic cells which may be transported through the venous vessels to the umbilicus.

However, when primary UE is associated with endometriosis at other, mainly pelvic, sites, the metastatic hypothesis appears also plausible. Of note, the prevalence of endometriotic pelvic lesions co-existent with primary UE was much higher than the usual estimates observed in the general premenopausal population [1,2,3,4]. This finding supports the possibility of a common etiological mechanism for the two disease locations because, if a separate pathogenesis exists for UE, the unusually high frequency of concomitant pelvic lesions would be difficult to explain.

If transtubal menstrual reflux is the origin of endometriosis, once in the pelvis endometrial cells might reach the umbilicus and implant directly on its parietal peritoneal surface without necessarily entering the vascular or lymphatic vessels. Indeed, a large body of evidence supports the notion that intra-abdominal endometriotic lesion distribution is determined to a large extent by physiological and anatomical factors that favor endometrial cell implantation [47].

Large bowel peristalsis combined with diaphragmatic respiratory movements, i.e., a physiologic factor, originate hydrostatic pressure variations that convey the peritoneal fluid from the pelvis along the right peritoneal gutter to the retro-hepatic and sub-phrenic area [48,49,50]. The falciform ligament, i.e., an anatomical factor, hampers the transit across the midline from the right to the left sub-phrenic space [51,52]. This explains the much higher prevalence of right-sided diaphragmatic, hepatic, and pleural endometriosis lesions compared with left-sided ones and could also elucidate the origin of UE.

In fact, once stuck by the falciform ligament, the peritoneal fluid containing free-floating endometriotic glands could be funneled by this crescentic peritoneal fold and by the hepatic round ligament toward the intra-abdominal aspect of the umbilicus, where endometriotic cells may eventually implant and infiltrate the overlying connective tissue. The presence of a small peritoneal concavity creates a sort of niche that would further facilitate endometrial cells implantation. This is also supported by reports of UE in women affected by umbilical hernia [27,40,53].

Even in the hypothesis of peritoneal fluid ascending directly from the pelvis along the anterior abdominal wall, endometrial cells would be channeled by the peritoneal folds created by the medial umbilical ligaments (obliterated umbilical arteries) and the median umbilical ligament (obliterated uracus), which create two anatomic preferential routes converging toward the umbilicus [54].

The pattern of dissemination of ovarian epithelial cancer cells might constitute a similar pathogenic model here. Both hematogenous and lymphatic spread have been suggested as the physio-pathological mechanism to explain umbilical metastases, the so-called Sister Mary Joseph’s nodules, from gynecologic cancers [10,55]. However, according to Hugen et al. (2021) [56], a local direct extension of cancer cells from the peritoneal aspect to the skin is possible, as the umbilicus is a deep structure directly connected to the extraperitoneal tissue.

Indeed, a contiguous extension of ovarian cancer cells from the peritoneal surface to the umbilical skin has been demonstrated [57]. Moreover, in a recent nationwide review of pathology records of umbilical metastases diagnosed in the Netherlands between 1979 and 2015, almost three fourths of the 806 study patients were females and, in these cases, umbilical metastases most frequently originated from the ovaries [56].

The peritoneal fluid conveyance hypothesis would be in line with evidence on endometriosis affecting the liver, diaphragm, and pleura [47,58,59], and would add further support to the unifying mechanistic theory of endometriosis as a disease originating from menstrual reflux and subsequent abdominopelvic distribution of lesions according to specific local characteristics that facilitates implantation [60]. If this is the case, primary UE should be considered part of the “right hypochondrium complex” [47]. Of note, this hypothesis would explain primary UE even in the absence of co-existent pelvic implants, as the origin of ectopic umbilical endometriotic glands would be the retrograde menstrual flow per se, independently of already extant pelvic localizations of the disease.

The sine qua non for the validity of this theory is the demonstration of a continuum of endometriotic infiltration through the entire thickness of the umbilical scar with involvement of the parietal peritoneal layer, as shown in Figure 2, Figure 3 and Figure 4.

Future studies on UE should focus on detailed histopathology findings of those women who undergo en-bloc omphalectomy. In addition, when laparoscopy is also indicated, a careful visual inspection of the diaphragm after liver retraction by means of a blunt probe should be performed to identify concomitant endometriotic implants [61]. These measures would both help clarify whether UE originates from inside the abdominal cavity or is a purely skin condition with a separate pathogenesis.

## Figures and Tables

**Figure 1 jcm-11-00995-f001:**
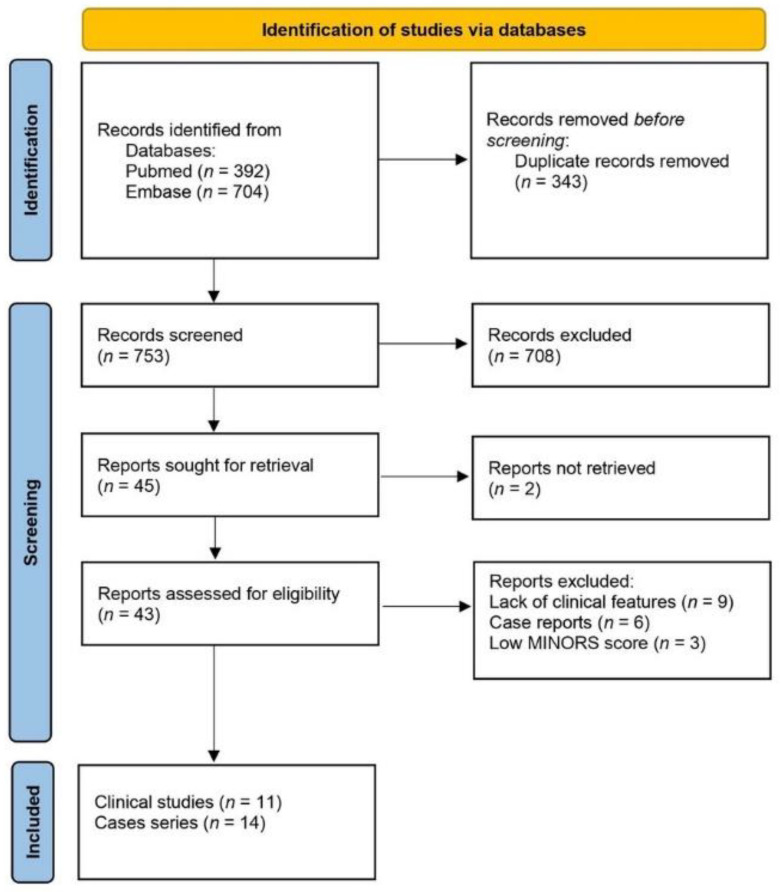
Flow diagram of study identification and selection.

**Figure 2 jcm-11-00995-f002:**
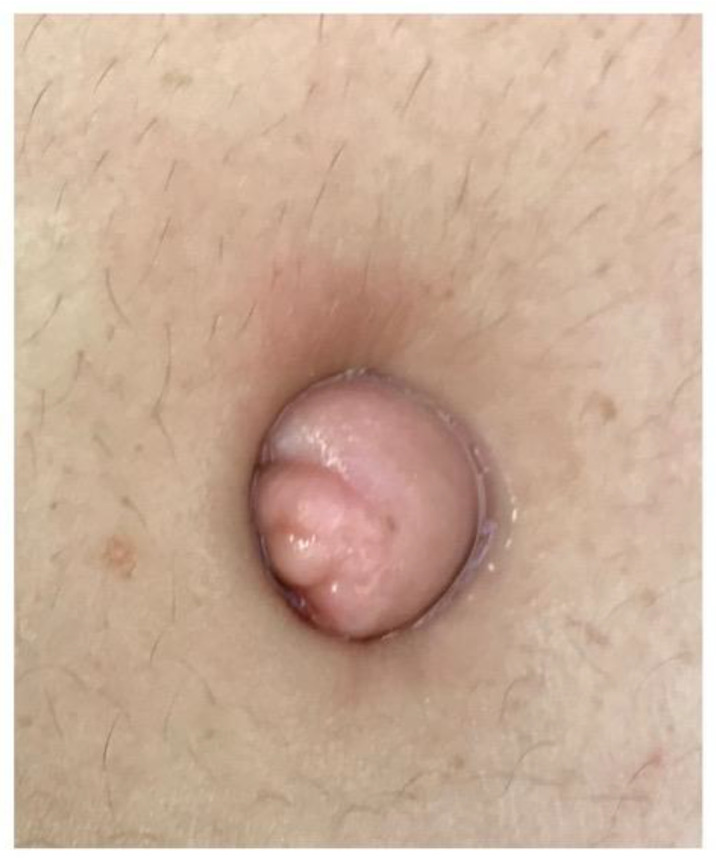
Endometriosis infiltrating the entire skin aspect of the umbilicus.

**Figure 3 jcm-11-00995-f003:**
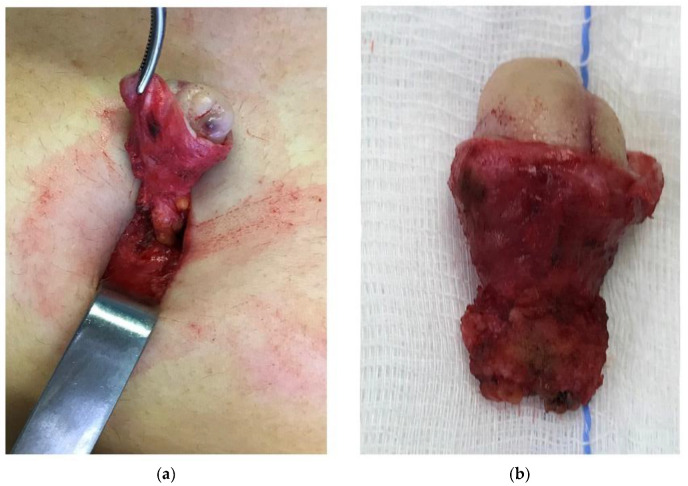
The umbilical peduncle is dissected down to the parietal peritoneum, which is included in the resected tissue (**a**). Anatomical specimen of en-bloc resection of umbilical endometriosis from the skin aspect to the parietal peritoneum (**b**).

**Figure 4 jcm-11-00995-f004:**
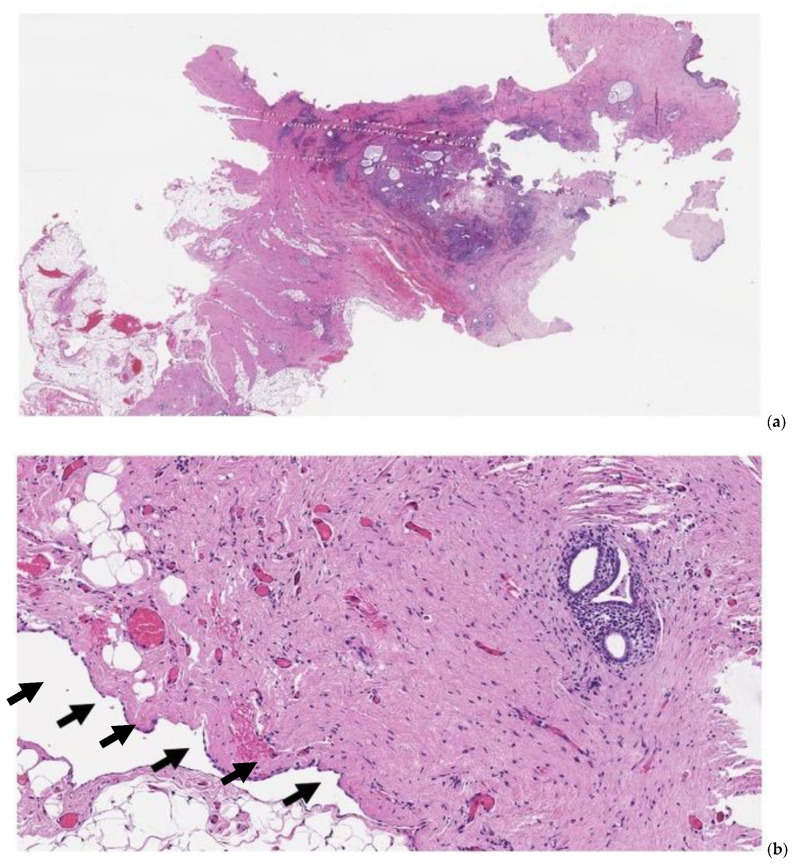
Pathology specimen after full-thickness omphalectomy for umbilical endometriosis. At scanning magnification ((**a**), hematoxylin-eosin, 10×), foci of endometriosis are apparent, spanning from the deeper tissues of the abdominal wall throughout the dermal layer (left to right). At higher magnification ((**b**), hematoxylin-eosin, 100×), endometrial glands are identified in the stroma underlying the parietal peritoneal mesothelial surface (arrows).

**Table 1 jcm-11-00995-t001:** Main characteristics of the selected studies.

Author, Year of Publication (Ref)	Country	Study Design ° Period of Recruitment °°	Women with Abdominal Wall Endometriosis (*n*)	Women with UE (*n*)	Age (Mean) of Women with UE	Primary UE % (*n*)	Secondary UE % (*n*)	MINORS Score
**Clinical** **series**								
Steck and Helwing, 1965 [31]	USA	Retrospective study on recorded cases	70	28	NR	75.0 (21/28)	25.0 (7/28)	8
McKenna and Wade-Evans, 1985 [30]	UK	Retrospective study on recorded cases between 1951–1980	25	5	47.5	100 (5/5)	0	8
Zhao et al., 2005 [29]	China	Retrospective study on recorded cases between 1951–1980	62	2	NR	100 (2/2)	0	10
Agarwal et al., 2008 [28]	Singapore	Retrospective study on recorded cases between 2000–2007	8	2	42	50.0 (1/2)	50.0 (1/2)	9
Leite et al., 2009 [27]	Brazil	Retrospective study 2001–2007	31	2	NR	0	100 (2/2)	10
Savelli et al., 2012 [26]	Italy	Retrospective study 2001–2007	19	8	NR	25.0 (2/8)	75.0 (6/8)	8
Ecker et al., 2014 [25]	USA	Retrospective study 2001–2013	63	9	NR	44.4 (4/9)	55.6 (5/9)	8
Vellido-Cotelo et al., 2015 [24]	Spain	Retrospective study 2000–2012	15	2	37	0	100 (2/2)	8
Chrysostomou et al., 2017 [32]	South Africa	Prospective study 2010–2016	14	6	31.1	100 (6/6)	0	13
Marras et al., 2019 [23]	Switzerland	Retrospective study 2007–2017	35	10	NR	60.0 (6/10)	40.0 (4/10)	11
Youssef, 2020 [22]	Egypt	Retrospective study 2016–2019	21	2	NR	50.0 (1/2)	50.0 (1/2) §	9
**Total clinical series**			363	76		63.2 (48/76)	36.8 (28/76)	
95% Confidence Interval						51.9–73.1	26.9–48.1	
Case series								
Rabinovitch et al., 1952 [46]	USA	NR		3	36	100 (3/3)	0	6
Pathak and Hayes, 1968 [33]	Jamaica	NR		3	42	66.7 (2/3)	33.3 (1/3)	10
Lattuneddu et al., 2002 [34]	Italy	2000–2001		3	43.6	33.3 (1/3)	66.7 (2/3)	9
Al-Saad et al., 2007 [35]	Kingdom of Bahrain	2002–2005		3	42.3	100 (3/3)	0	11
Dessy et al., 2008 [36]	Italy	2003–2006		4	37	50.0 (2/4)	50.0 (2/4)	11
Fedele et al., 2010 [37]	Italy	2001–2003		7	37	57.1 (4/7)	42.9 (3/7)	9
Abramowicz et al., 2011 [38]	France	NR		3	29	100 (3/3)	0	9
Darouichi et al., 2013 [39]	Switzerland	NR		3	37.3	66.7 (2/3)	33.3 (1/3)	7
Saito et al., 2013 [40]	Japan	1999–2011		7	35.7	71.4 (5/7)	28.6 (2/7)	11
Chikazawa et al., 2014 [41]	Japan	NR		3	37	33.3 (1/3)	66.7 (2/3)	7
Boesgaard-Kjer et al., 2017 [42]	Denmark	2011–2014		10	28.5	100 (10/10)	0	9
Dos Santos Filho et al., 2018 [43]	Brazil	2014–2017		6	33	100 (6/6)	0	11
Hirata et al., 2020 [9]	Japan	2006–2016		96	40	66.7 (64/96) *	32.3 (31/96) *	11
Makena et al., 2020 [44]	Kenya	2015–2019		5	40	80.0 (4/5)	20.0 (1/5)	10
**Total case series**				156		71.0 (110/155)	29.0 (45/155)	
95% Confidence Interval						63.4–77.5	22.5–36.6	
**Total all studies**				232		68.4 (158/231)	31.6 (73/231)	
95% Confidence Interval						62.1–74.1	26.0–37.9	

Legend: ° for clinical series; °° if available; NR: not reported; § woman with a previous hysterosonography; * one patient had unknown surgical history.

**Table 2 jcm-11-00995-t002:** Reproductive and endometriosis history of women with UE.

Author, Year of Publication (Ref)	Parous (Parae/Total Women with UE) % (*n*)	Cesarean Section (Women with a History of CS/Total Parae) % (*n*)	Previous Abdominal Surgery (Women with Previous Abdominal Surgery/Total Cases with UE) % (*n*)	History of Endometriosis (Women with History of Endometriosis/Total Cases UE) % (*n*)	If Laparoscopy Associated, Concomitant Endometriosis % (*n*)
**Clinical** **series**					
Steck and Helwing, 1965 [31]	NR	NR	25.0 (7/28)	NR	NR
McKenna and Wade-Evans, 1985 [30]	100 (5/5)	0	0	40.0 (2/5)	20.0 (1/5)
Zhao et al., 2005 [29]	NR	NR	0	NR	NR
Agarwal et al., 2008 [28]	100 (2/2)	50.0 (1/2)	50.0 (1/2)	0	50.0 (1/2)
Leite et al., 2009 [27]	100 (2/2)	100 (2/2)	100 (2/2)	NR	NR
Savelli et al., 2012 [26]	NR	NR	75.0 (6/8)	75.0 (6/8)	NR
Ecker et al., 2014 [25]	33.3 (3/9)	33.3 (1/3)	55.5 (5/9)	NR	NR
Vellido-Cotelo et al., 2015 [24]	50.0 (1/2)	100 (1/1)	100 (2/2)	50 (1/2)	NR
Chrysostomou et al., 2017 [32]	NR	NR	0	0	0
Marras et al., 2019 [23]	NR	NR	40.0 (4/10)	60.0 (6/10)	80.0 (8/10)
Youssef, 2020 [22]	NR	NR	50.0 (1/2)	NR	50.0 (1/2)
**Total clinical series**	65.0 (13/20)	38.5 (5/13)	36.8 (28/76)	45.5 (15/33)	44.0 (11/25)
95% Confidence Interval	43.3–81.9	17.7–64.5	26.9–48.1	29.8–62.0	26.7–62.9
**Case series**					
Rabinovitch et al., 1952 [46]	NR	NR	0	NR	NR
Pathak and Hayes, 1968 [33]	0	0	33.3 (1/3)	NR	33.3 (1/3) §
Lattuneddu et al., 2002 [34]	33.3 (1/3)	100 (1/1)	33.3 (1/3)	NR	33.3 (1/3)
Al-Saad et al., 2007 [35]	66.7 (2/3)	0	0	NR	NR
Dessy et al., 2008 [36]	75.0 (3/4)	66.7 (2/3)	50.0 (2/4)	0	NR
Fedele et al., 2010 [37]	14.3 (1/7)	NR	42.8 (3/7)	71.4 (5/7)	28.6 (2/7)
Abramowicz et al., 2011 [38]	NR	NR	0	100 (3/3)	100 (3/3)
Darouichi et al., 2013 [39]	66.7 (2/3)	50.0 (1/2)	33.3 (1/3)	33.3 (1/3)	33.3 (1/3)
Saito et al., 2013 [40]	28.6 (2/7)	NR	28.6 (2/7)	14.3 (1/7)	14.3 (1/7)
Chikazawa et al., 2014 [41]	100 (3/3)	66.7 (2/3)	66.7 (2/3)	0	33.3 (1/3)
Boesgaard-Kjer et al., 2017 [42]	20.0 (2/10)	0	0	NR	10.0 (1/10)
Dos Santos Filho et al., 2018 [43]	100 (6/6)	0	0	0	NR
Hirata et al., 2020 [9]	63.5 (61/96)	18.0 (11/61)	32.3 (31/96) °	NR	NR
Makena et al., 2020 [44]	60.0 (3/5)	33.3 (1/3)	20.0 (1/5)	NR	40.0 (2/5)
**Total case series**	57.3 (86/150)	22.2 (18/81)	28.2 (44/156)	30.3 (10/33)	29.5 (13/44)
95% Confidence Interval	49.3–65.0	14.5–32.4	21.7–35.7	17.4–47.3	18.2–44.2
**Total all studies**	58.2 (99/170)	24.5 (23/94)	31.0 (72/232)	37.9 (25/66)	34.8 (24/69)
95% Confidence Interval	50.7–65.4	16.9–34.1	25.4–37.3	27.2–49.9	24.6–46.6

Legend: * Patients with previous pelvic surgery or caesarean section who presented with scar or umbilical endometriosis were excluded from the study; § woman with adenomyosis; ° one patient had unknown surgical history.

**Table 3 jcm-11-00995-t003:** Clinical symptoms of UE.

Author, Year of Publication (Ref)	Pain % (*n*)	Bleeding % (*n*)	Catamenial Symptoms * % (*n*)
**Clinical** **series**			
Steck and Helwing, 1965 [31]	NR	NR	NR
McKenna and Wade-Evans, 1985 [30]	NR	NR	80.0 (4/5)
Zhao et al., 2005 [29]	NR	NR	NR
Agarwal et al., 2008 [28]	50.0 (1/2)	100 (2/2)	100 (1/2)
Leite et al., 2009 [27]	NR	50.0 (1/2)	NR
Savelli et al., 2012 [26]	NR	NR	NR
Ecker et al., 2014 [25]	55.5 (5/9)	NR	NR
Vellido-Cotelo et al., 2015 [24]	NR	NR	100 (2/2)
Chrysostomou et al., 2017 [32]	100 (6/6)	50.0 (3/6)	0
Marras et al., 2019 [23]	NR	40.0 (4/10)	NR
Youssef, 2020 [22]	100 (2/2)	50.0 (1/2)	100 (2/2)
**Total clinical series**	73.7 (14/19)	50.0 (11/22)	52.9 (9/17)
95% Confidence Interval	51.2–88.2	30.7–69.3	31.0–73.8
**Case series**			
Rabinovitch et al., 1952 [46]	100 (3/3)	0 (0/3)	100 (3/3)
Pathak and Hayes, 1968 [33]	100 (3/3)	0 (0/3)	66.6 (2/3)
Lattuneddu et al., 2002 [34]	100 (3/3)	NR	NR
Al-Saad et al., 2007 [35]	100 (3/3)	100 (3/3)	100 (3/3)
Dessy et al., 2008 [36]	50.0 (2/4)	25.0 (1/4)	100 (4/4)
Fedele et al., 2010 [37]	100 (7/7)	57.0 (4/7)	100 (7/7)
Abramowicz et al., 2011 [38]	66.6 (2/3)	33.3 (1/3)	66.6 (2/3)
Darouichi et al., 2013 [39]	66.6 (2/3)	33.3 (1/3)	33.3 (1/3)
Saito et al., 2013 [40]	100 (7/7)	57.0 (4/7)	86.0 (6/7)
Chikazawa et al., 2014 [41]	100 (3/3)	0 (0/3)	33.3 (1/3)
Boesgaard-Kjer et al., 2017 [42]	NR	100 (10/10)	100 (10/10)
Dos Santos Filho et al., 2018 [43]	100 (6/6)	100 (6/6)	100 (6/6)
Hirata et al., 2020 [9]	81.0 (78/96)	44.8 (43/96)	86.5 (83/96)
Makena et al., 2020 [44]	80.0 (4/5)	100 (5/5)	100 (5/5)
**Total case series**	84.2 (123/146)	51.0 (78/153)	86.9 (133/153)
95% Confidence Interval	77.5–89.3	43.1–58.8	80.7–91.4
**Total all studies**	83.0 (137/165)	50.9 (89/175)	83.5 (142/170)
95% Confidence Interval	76.6–88.0	43.5–58.2	77.2–88.4

Legend: * catamenial symptoms: swelling, color change, consistency change, and tenesmus.

**Table 4 jcm-11-00995-t004:** Frequency of recurrence of UE.

Author, Year of Publication (Ref)	Cases of UE (*n*)	Treatment	Duration of Follow Up (Mean in Months and Range)	Recurrence % (*n*)	Time to Recurrence (Mean in Months)
**Clinical series**					
Agarwal et al., 2008 [28]	2	surgery	20.5	0	
Leite et al., 2009 [27]	2	surgery	NR	100 (2/2)	NR
Chrysostomou et al., 2017 [32]	6	surgery	36 (6–72)	0	
Marras et al., 2019 [23]	10	surgery	62.4 ± 39.6 §	10.0(1/10)	24
**Case series**					
Pathak and Hayes, 1968 [33]	3	surgery	24 (2 cases) NR (1 case)	33.3 (1/3)	22
Lattuneddu et al., 2002 [34]	3	surgery	24	0	
Al-Saad et al., 2007 [35]	3	surgery	28	0	
Dessy et al., 2008 [36]	4	surgery	13	0	
Fedele et al., 2010 [37]	7	surgery	92.5	0	
Abramowicz et al., 2011 [38]	3	surgery	3	0	
Saito et al., 2013 [40]	7	1 surgery **	58.3	0	
Dos Santos Filho et al., 2018 [43]	6	surgery	NR	0	
Hirata et al., 2020 [9]	87 *	surgery	6870	3.4 (3/87)	Recurrence occurred in 3 women, at 3, 8, and 12 months after excision without peritoneum. °
Makena et al., 2020 [44]	5	surgery	NR	0	
**Total**	148			7	

Legend: § information based on all the cases of abdominal wall endometriosis; * 9 cases are not considered in the analysis of recurrence; ** 1/7 underwent radical resection of umbilical nodule, 2/7 underwent expectant management, 4/7 took estrogen-progestogen hormone therapy or Dienogest; ° % cumulative recurrence rate: 6 months—1.34, 12 months—6.35, 60 months—6.35.

## Data Availability

All data generated or analyzed during this study are included in this published article.

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
