# Peer review of "Umbilical Endometriosis: A Systematic Literature Review and Pathogenic Theory Proposal"

_jcm, 2022, doi:10.3390/jcm11040995_

Round 1

Reviewer 1 Report

This paper is a systematic literature review of umbilical endometriosis, and I think it is very useful to hypothesize the pathogenic theory of this disease from the accumulation of previous papers. It is possible to accept it as it is, but it will be a better review if there is a summary and consideration of the effect of conservative treatments (hormonal treatments) on this disease.

For example, while it is recommended to add hormone therapy such as OC and dienogest to prevent postoperative recurrence of endometriosis (endometriotic cyst), readers will want to know how about umbilical endometriosis has been reported.

Author Response

Response to Reviewer 1 Comments

Point 1: I This paper is a systematic literature review of umbilical endometriosis, and I think it is very useful to hypothesize the pathogenic theory of this disease from the accumulation of previous papers. It is possible to accept it as it is, but it will be a better review if there is a summary and consideration of the effect of conservative treatments (hormonal treatments) on this disease.

For example, while it is recommended to add hormone therapy such as OC and dienogest to prevent postoperative recurrence of endometriosis (endometriotic cyst), readers will want to know how about umbilical endometriosis has been reported.

Response 1: We thank the reviewer for the comment. The review is focused on the epidemiological and pathogenic aspects of umbilical endometriosis; therefore, we did not deeply deal with the management of the disease. Accordingly with the reviewer comment, we add a paragraph on clinical management (lines 69-75 and lines 185-186). In addition, we have introduced in table 4 the type of treatment carried out, and we have indicated in which case hormonal treatments were administered. However, the limited number of information does not allow an assessment of the role of hormonal therapy on the risk of relapse.

Reviewer 2 Report

Reviewer

Comments and Suggestions for Authors

This study attempts to evaluate umbilical endometriosis and present a possible pathogenic theory even though there are relatively few articles in the literature.

The topic is interesting for clinicians and current, but the manuscript needs major revisions.

  • Because a possible UE pathogenic theory is proposed, the postulated and currently favored theories should be presented in the Introduction.

  • Line 62-66 vs line 20 – Clinical management is missing.

  • It is interesting to introduce the size of the umbilical nodule in table 4 presented.

  • There is any correlation between the size of the umbilical nodule and the recurrence rate? There was no recurrence after wide resection including of the peritoneum (Hirata et al).

  • Another possible theory - Romera-Barba et al. demonstrated that the disease occurs after prolonged exposure to the metaplastic and environmental factors that catalyze the development of umbilical endometriosis.

  • In Discussions

  • In this study, it was possible to establish the role of imaging diagnosis (the MRI/ultrasound features of umbilical endometriosis) and planning of management for abdominal wall reconstruction.
  • Differential diagnoses of umbilical endometriosis is another possible discussion goal
  • The clinical management of these cases was not presented in light of the results obtained in this review.

Author Response

Response to Reviewer 2 Comments

Point 1: This study attempts to evaluate umbilical endometriosis and present a possible pathogenic theory even though there are relatively few articles in the literature.

The topic is interesting for clinicians and current, but the manuscript needs major revisions.

Because a possible UE pathogenic theory is proposed, the postulated and currently favored theories should be presented in the Introduction.

Response 1: We thank the reviewer for the comment. A paragraph on postulated and current theories has been added in the introduction section (lines 55-59).

Point 2: Line 62-66 vs line 20 – Clinical management is missing.

Response 2: In order to maintain the review updated with the most recent evidence, a short paragraph has been added with the clinical management of umbilical endometriosis (lines 68-74).

Point 3: It is interesting to introduce the size of the umbilical nodule in table 4 presented.

There is any correlation between the size of the umbilical nodule and the recurrence rate? There was no recurrence after wide resection including of the peritoneum (Hirata et al).

Response 3: A total of 12 case series and only 2 clinical series reported data on lesion size, but they do not report data on the correlation between size and the risk of recurrence.

Only one Clinical series, Zaho et al., is not included in table 4 because it reports data on the risk of recurrence in relation to the size of the endometrial nodules of the abdominal wall in general without specifying data on the umbilical location.

We have added the lesion size data (lines 178-180). As suggested by the reviewer, we have also included the consideration of Hirata et al. (lines 185-187).

Point 4: Another possible theory - Romera-Barba et al. demonstrated that the disease occurs after prolonged exposure to the metaplastic and environmental factors that catalyze the development of umbilical endometriosis.

Response 4: We thank the reviewer for the advice, we have included in the introduction the theory proposed by Omera-Barba et al. (lines 63-65).

Point 5: In this study, it was possible to establish the role of imaging diagnosis (the MRI/ultrasound features of umbilical endometriosis) and planning of management for abdominal wall reconstruction.

Differential diagnoses of umbilical endometriosis is another possible discussion goal

The clinical management of these cases was not presented in light of the results obtained in this review.

Response 5: Our review explicitly highlights the epidemiological aspects and clarifies the possible etiopathogenetic cases of umbilical endometriosis. Considering the scarcity of data present in the literature and their heterogeneity, we have not dealt with the diagnostic and treatment aspects. The management for abdominal wall reconstruction is reported in figure 3.

Round 2

Reviewer 2 Report

Dear Authors,
Your article adds value to a topic as publicized as endometriosis.
A single mention on line 57 - instead of severe, several will be mentioned.

Kind regards